# Effects of Road Slope and Vehicle Weight on Truck Fuel Consumption

**John Jairo Posada-Henao** [1,*] **, Iván Sarmiento-Ordosgoitia** [1] **and Alexánder A. Correa-Espinal** [2]

1    Department of Civil Engineering, Facultad de Minas, Universidad Nacional de Colombia at Medellín, Medellín 050034, Colombia

2    Department of Organization Engineering, Facultad de Minas, Universidad Nacional de Colombia at Medellín, Medellín 050034, Colombia

*    Correspondence: jjposada@unal.edu.co; Tel.: +57-6044255150

**Abstract:** In this paper, we developed truck fuel consumption models for the particular assistance of professionals in charge of road project valuation in terms of predicting fuel used by trucks, which is an important topic on vehicle operating costs to be considered in the benefit–cost analysis of road projects. On the other hand, fuel consumption has a direct impact on emissions to the atmosphere, and thus future research can be conducted regarding estimations about emissions by trucks. In this research, we identified the effect of overall vehicle weight on truck fuel consumption in a free-flow regime. The methodology includes the design of experiments and factorial design as statistical techniques to obtain data, as well as linear and non-linear regressions to obtain models for two types of trucks: rigid (three axles) and articulated (six axles). Notably, there is no evidence of research previously conducted on the latter. We used statistical methods for the selection of trucks, equipment, road segments, and other aspects, obtaining good control in tests verifying the appropriate values for factors according to the planned ones. The results satisfy the expectations of the research, and it was demonstrated that the vehicle weight and roadway slope were significantly more important than speed alone, which was typically considered the main variable in other studies. On the other hand, longitudinal slopes higher than 5% were found to not be suitable for freight road corridors. It is recommended that 6-axle trucks instead of 3-axle trucks be used for a 16 t amount of cargo transported on a plain road (longitudinal slope under 3%). The HDM-4 model did not represent fuel consumption adequately for the current vehicle fleet operating on roads. Fuel consumption models must be updated, for instance, every 10 years, such that they can adapt to vehicle technological advances and the energetic improvement of fuels, including the proportion of biofuels and gas.

**Keywords:** freight transportation; truck fuel consumption; vehicle weight; road slope; truck speed; experiment design



## 1. Introduction

Road freight transportation represents the main alternative for mobilization of goods in many countries [1]. As such, an optimal road infrastructure is important in keeping the cost of transportation as low as possible in order to maintain an economic and competitive edge [2]. Fuel consumption has been studied since the advent of motor vehicles in the twentieth century. Interest on the topic has increased in recent years due to fuel costs as well as the economic and environmental impacts generated through its use, specifically in light of global warming phenomena and emissions, as well as due to it being a non-renewable resource [3–6]. In accordance with this, emission and fuel consumption typically increase with the increment of vehicle average speed [7], as can be seen in Figure 1 for $CO_2$ emissions and for two types of vehicles (light-duty gasoline and heavy-duty diesel).

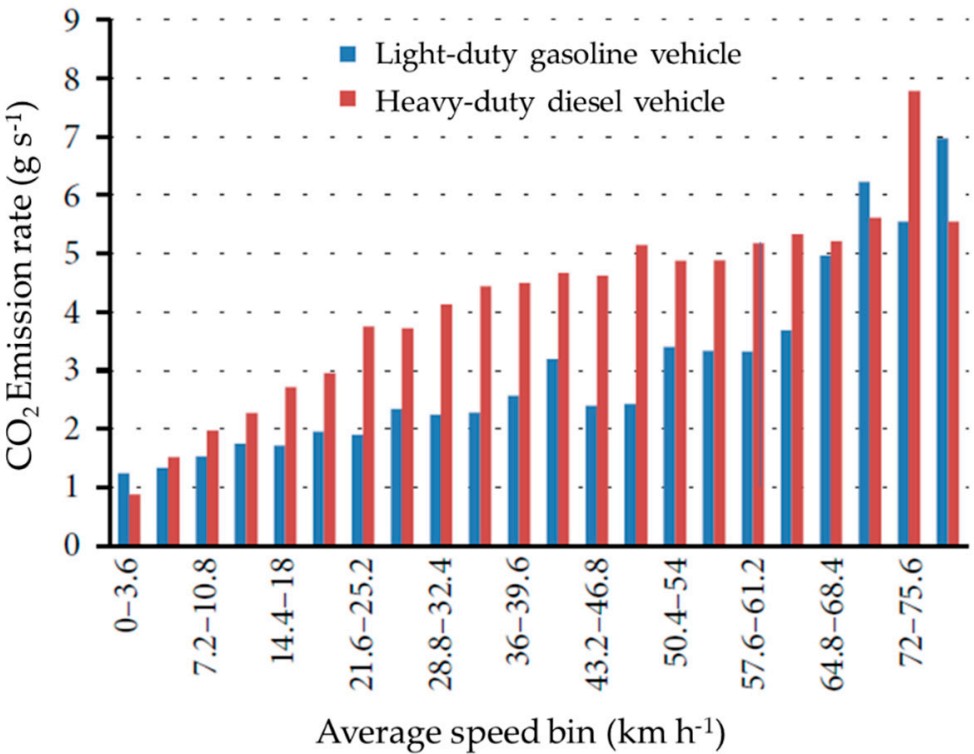

**Figure 1.** $CO_2$ emission prediction values under different average speed-bins, adapted from [7].

From Figure 1, it is noted that emission and fuel consumption of heavy-duty diesel vehicles is greater than of the light-duty gasoline vehicles, and that for the lower average speeds, there is a higher increasing for emission and fuel consumption rates, while those rates rise slowly when average speed increases.

Figure 2 shows $CO_2$ emissions and fuel consumption predicted according to vehicle speed for heavy-duty vehicles, and it can be seen that for lower and higher speeds, those emissions and fuel consumption increasing rates are greater than for intermediate speeds.

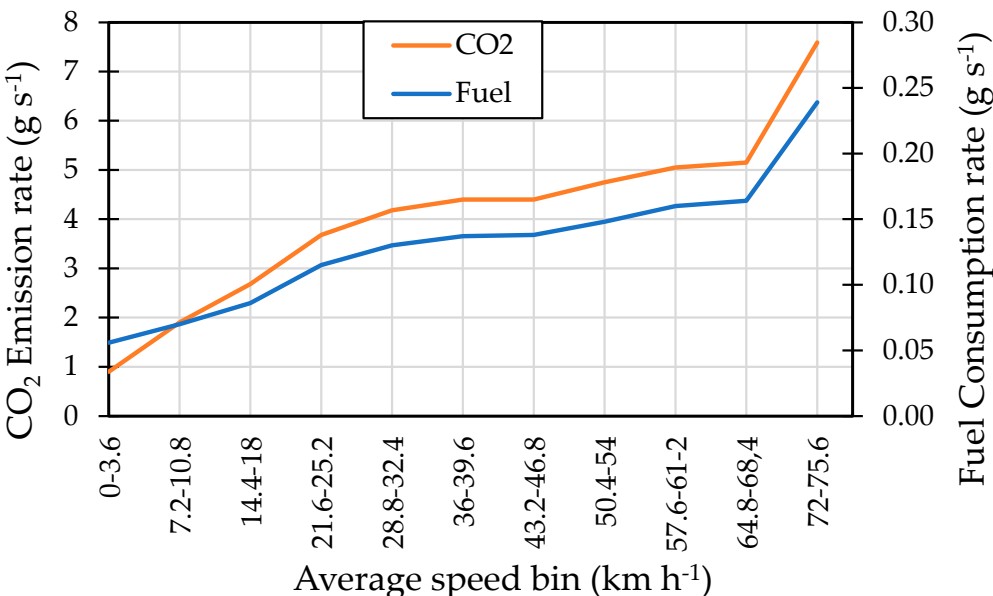

**Figure 2.** $CO_2$ emission and fuel consumption prediction values under different average speed-bins for heavy-duty vehicles. Proper elaboration using information from [7].

In this paper, we considered fuel consumption in trucks that use diesel fuel, which is a non-renewable fossil fuel. Its consumption and behavior under different operating conditions should be rightly known, due to the important role of fuel in determining transportation costs. Truck-type vehicles were studied, as their fuel consumption is higher than that of other vehicle types, making them important due to the associated impact on transportation costs, and as they have great impacts at both the economy level (as they mobilize freight for all of society) and, in a special way, at the country level (whose economy is dependent on road transport of its goods) they thus represent an important and essential link for the economic and social development of any country [8]. However, although many studies have focused on fuel consumption [3,9,10], some factors that affect it have not been studied with proper rigor [8,10,11], especially for trucks, as noted in this paper. As an example, the congestion due to continuous stops and starts and greater variability in speed could increase the fuel consumption by at least 25%, but this is very variable according to vehicle type, road condition, and congestion severity. On the other hand, aggressive driving impacts fuel consumption between 5% and 30% [12,13].

Other factors are as follows: fuel quality, vehicle maintenance, the amount of cargo carried by the truck (or, more broadly, the total weight of the vehicle), and the slope of the road at high values, all being aspects of great importance in fuel consumption, and they are also very normal conditions in cargo transportation since trucks are not always used with the same amount of cargo, or total weight, and many regions of countries have roads with variable longitudinal slope adopting high values due to topographic conditions [8,14].

This research addresses some of these gaps mentioned before, measuring and modelling the influence of two variables on fuel consumption: the overall truck weight and longitudinal slopes (especially when it takes high values). The amount of freight being carried by a vehicle is important, as it affects the power required by the vehicle and, thus, fuel consumption [7,10,14,15]. In the same way, fuel consumption depends by road longitudinal slope due the incremental requirement of power engine if the slope increase [14–20].

The scope considers two types of trucks: rigid and articulated (with three and six axles, respectively). A literature review yielded very little prior research on these variables regarding trucks with three axles, and there was no evidence of studies on the subject for the second type of truck (six axles), providing a situation in which to enable additional contributions to knowledge on fuel consumption. Furthermore, the better determination of fuel consumption allows for better estimates of pollutant emissions and their effect on the environment.

This research was conducted in a region that has appropriate traffic flow conditions, a range of longitudinal road slopes, and access to services to control the desired conditions for driving the trucks. The design of the experiments was used to study the data and produce model results; in particular, $3^K$ and $2^K$ factorial design techniques were considered the most appropriate for this study, as is explained later.

The results of this study have many benefits for researchers, who can adopt the methodology used or the novel models for fuel consumption in future studies, as well as to road designers, transportation projects valuators, transport carriers, and the government agencies responsible for the development of the construction and operation of the road network. From a proactive perspective, the findings of this study and the models developed herein are important to stakeholders, who can now assess their road projects using a realistic valuation of truck fuel consumption, providing an additional measure when comparing road design alternatives. This paper includes five sections: First, this introduction, followed by the theoretical framework for fuel consumption and related models. Next, the research methodology is presented; the results and analysis are detailed; and, finally, our conclusions are given.

## 2. Fuel Consumption by Vehicles

Here, we present a theoretical framework for fuel consumption by vehicles, as well as models to estimate such consumption and their evolution.

*2.1. Theoretical Framework*

The evaluation of a road project includes important variables, such as construction costs; property costs; value of travel time for users and goods; maintenance costs of the road; cost of vehicle operation; and other transportation-related costs as externalities, such as environmental impacts and others. Vehicle operation cost is the largest component in the total cost of transportation [11] and, further, fuel consumption is one of the most important components of vehicle operating costs in the salary of drivers. Studies have shown that fuel consumption, within total operating costs, comprises 33% for light vehicles [3], 50% for some heavy vehicles [9,21], and from 40 to 60% for trucks [22], while others have estimated it at 30% for all vehicles in general [4]. This depends on the region of the valuation, as well as the specific user time and fuel costs. For example, in Venezuela in 2001, fuel consumption was 4% of the vehicle operating costs, while it was 29% in Spain, 28% in Bolivia, and 24% in Colombia [23]; on the other hand, in Colombia, it accounted for about 33% of the operating costs for trucks in 2011 (including tolls, or 38% not including tolls), while at the beginning of 2021, it was 35% (or 40%, respectively) [24].

The energy sources for vehicles are in transition (for example, to electricity) due to emissions from fossil fuels and climate change. Some studies have been conducted on this topic [25], but fossil fuels are expected to be used for several years yet; as such, study and estimates relating to such fuels are still necessary. Those studies reveal that battery health affects the performance of battery electric buses. However, there is little attention to the effects of power matching and seasonality (which significantly affect the battery performances) on planning for charging infrastructure. The main result from the source is that there are significant performance differences regarding vehicle scheduling and charging among different bus fleets in the battery-electric-bus-based transit system, providing strong evidence about the necessity to consider powering match and the seasonality in the bus charging infrastructure layout.

In an economical evaluation study, tolls are generally considered an internal transfer of the system analyzed, and not a cost in themselves.

Decisions about what vehicle should be used for any specific trip or how a vehicular fleet should be composed, in terms of several types of vehicles and according to the type of trips to be done, must consider fuel costs, routing, travelling distance, and loading/unloading operations [26]; in this context, the estimation of fuel consumption is important to carry out, as it can help to obtain better decisions.

Predicting fuel consumption is difficult due to the number of influencing parameters, as well as the technological advances in vehicles and in fuel types [27]; however, these advances do not necessarily lead to the expected reduction in fuel consumption, as vehicles require more power for the increased amenities offered [6], including air conditioning, refrigeration, and load capacity, among others. However, governments have been striving to increase fuel and vehicle efficiency to reduce fuel consumption in order to obtain economic and environmental improvements [6,28]. Freight companies and truckers are willing to pay for these new technologies, as they will recover their investment over the period of vehicle use [6]. The latter has been validated, considering that vehicle characteristics are important for fuel consumption [16], and that fuel costs affect driving patterns and, therefore, fuel consumption [29]; in this sense, eco-driving and eco-routing appear as good practices [20,30].

The amount of freight being carried by a vehicle is important, as it affects the power required by the vehicle and, thus, fuel consumption. Almost 30% of trucks on roads travel empty [31,32], while 20% travel empty in urban areas [33]. In other situations, such as that studied in Colombia, 12% of the trucks were found to operate overweight [34]. Trucks in general do not use their capacity to its limits (small load factor). For instance, in Colombia from 1997 to 2004, freight vehicles showed an average use of 50% by weight, and 74% by volume [2]. This pre-supposes an oversupply to be analyzed by type of truck and product transported, as some products (e.g., fuel, refrigerated products, and coal) require special transportation, implying close to 100% empty returning trips. Empty trips are normal

in the freight transportation industry, and they occur for various reasons, including load demand asymmetry between origins and destinations, imports/exports, products without return freight, and production periods; hence, oversupply cannot be considered a problem only for the weight average use. Furthermore, current logistics govern flow sizes, and under-utilization of vehicle space must not necessarily be interpreted as a lack of efficiency in economic activity.

*2.2. Previous Models to Estimate Fuel Consumption*

Studies and research in various parts of the world have developed models to predict fuel consumption under different vehicle types (passenger cars, buses, and light and heavy trucks, but not 6-axle articulated trucks) and use conditions. The most recent—and most reliable—models are of the mechanistic type (i.e., they mathematically determine and model the physics of the phenomenon under study, on the basis of real data and the behavior of the variables considered; statistical models only consider actual data), which can be adapted and calibrated to be used in conditions different from those in which they were developed [6,9,15,35,36].

Assuming default values in models possibly leads to prediction errors and, in the case of estimating fuel consumption, differences of up to 200% have been obtained with respect to the real values. The greatest differences are for trucks in congested traffic [10]. This evidence, along with issues such as obsolescence, technology, driver behaviors, and transportation policies force the calibration of vehicle operation cost and fuel consumption models to better reflect local conditions [9,21].

Recent research has been centered on studying the effect of aspects such as speed, road geometry, and pavement type and condition (among others) on fuel consumption. In addition, the available models have been developed on the basis of averages of results obtained in different studies [37], using new vehicles [21], and thus do not correspond to the conditions of specific locations or vehicle fleet age.

A first approach to modelling vehicle operation costs considers, for each component (including fuel), the characteristics of the road including an error term, according to Equation (1) [21]:

$$c = x \times f + e, \tag{1}$$

where

$c$ = cost or consumption of element, sometimes replaced by $\log(c)$ to facilitate obtaining results in a linear manner;
$x$ = vector of road characteristics;
$f$ = vector of coefficient, usually determined by ordinary least squares;
$e$ = error.

Other studies have proposed that fuel consumption is a function of vehicle speed, as in Equation (2) [21]. According to this equation, fuel consumption behavior reflects a "U"-shaped curve, as can be seen in Figure 3, indicating that there is high fuel consumption at relatively high or low speed, and minimum values at a certain speed [21].

$$F = a + b/V + c * V^2, \tag{2}$$

where

$F$ = fuel consumption per unit of distance (lt km$^{-1}$);
$V$ = vehicle speed (km h$^{-1}$);
$a$ = coefficient to include the effects of road characteristics (geometric and surface roughness) and vehicle (gross weight and power to weight ratio);
$b$ and $c$ = coefficients regarding geometric road characteristics (especially its longitudinal slope).

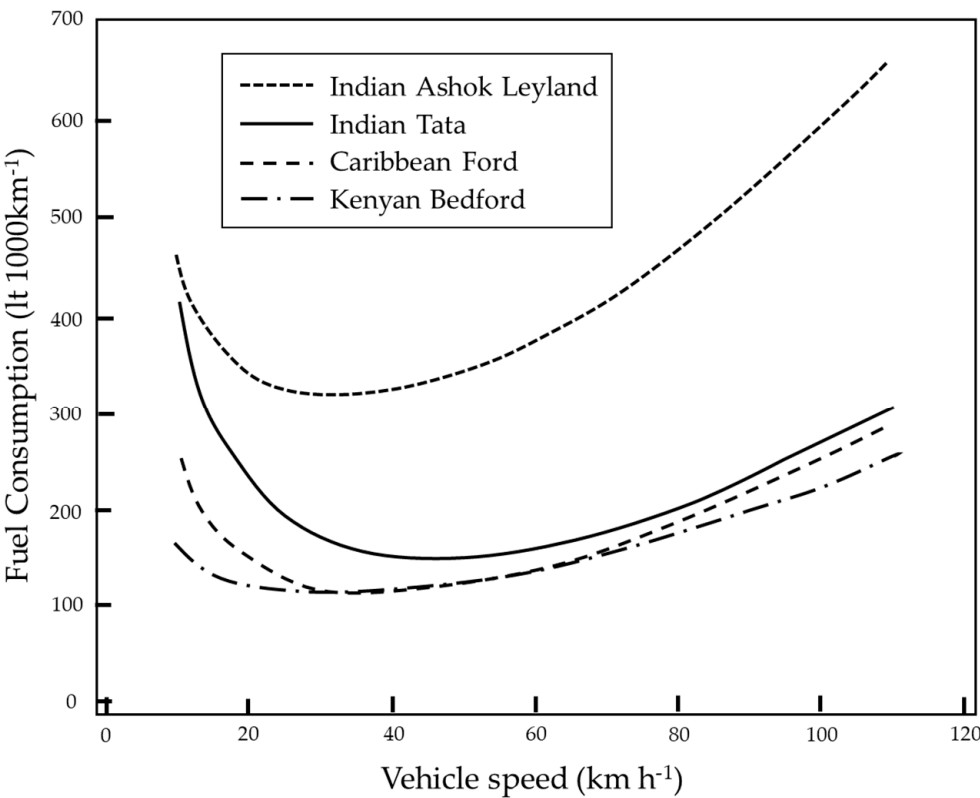

**Figure 3.** Vehicle consumption vs. vehicle speed for trucks. Redrawn from Chesher and Harrison [21].

These coefficients are positive and have been defined according to experimental studies.

As per a laboratory study, the proportionality between consumption and vehicle speed up to a power of three has also been proposed [27] with behavior, as depicted in Figure 3.

A study in Brazil visually considered truck load (empty, half full, and fully loaded) and defined different expressions for ascending and descending slopes, as well as including road curvature, in the case of buses [21]. The development of such models has been made evident with the inclusion of road slope (ascending/descending) and pavement condition into the International Regularity Index (IRI) [11].

Other mathematical models have specifically been developed for heavy vehicles, including variables such as speed (with various exponent values) and considering different slopes (ascending, descending, or null) in linear and non-linear models [3]. The fuel consumption and exhaust emissions have been shown to have a significant dependence on the road slope [20].

A widely applicable model is the Highway Development and Management-4 (HDM-4) [9,35,36], due to the extensive research behind its development. It has been used in nearly 100 countries under various conditions [37]. It is a mechanistic model, enabling the easy modelling of several vehicles and road characteristics, as well as changes through technological improvements [18]. The HDM-4 is the latest version of the HDM model, developed by the World Bank and others to plan and program investments in roads through simulating various alternatives over the analysis period [8].

The application of any model must include various factors that affect vehicle operating costs (VOC), and such models require calibration. For instance, Figure 4 shows vehicle operation costs for trucks in Canada, measured and predicted with the HDM using default values [37]. Differences can be observed in the total vehicle operating costs (VOC) and in the participation of the different components, suggesting that model calibration is necessary to ensure more accurate prediction results [38]. Another example of the need to calibrate and adapt models—especially fuel consumption models—is shown in Figure 5, in which the differences observed in several studies are presented. The only match is the shape of

the curves indicating high fuel consumption at low and high speeds, as well as minimum consumption at intermediate speed (as in Figure 3).

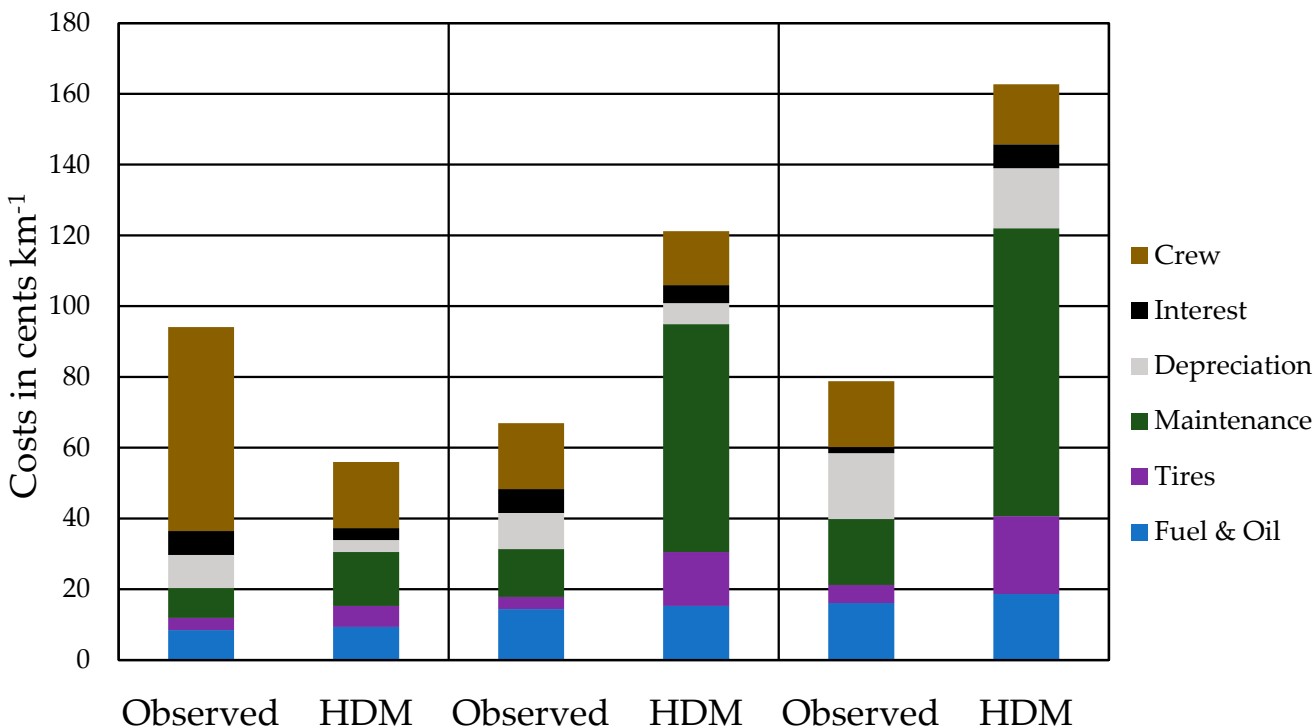

**Figure 4.** Comparison between observed and predicted VOC using non-calibrated HDM for trucks in Canada. Redrawn from Bennett and Paterson [37].

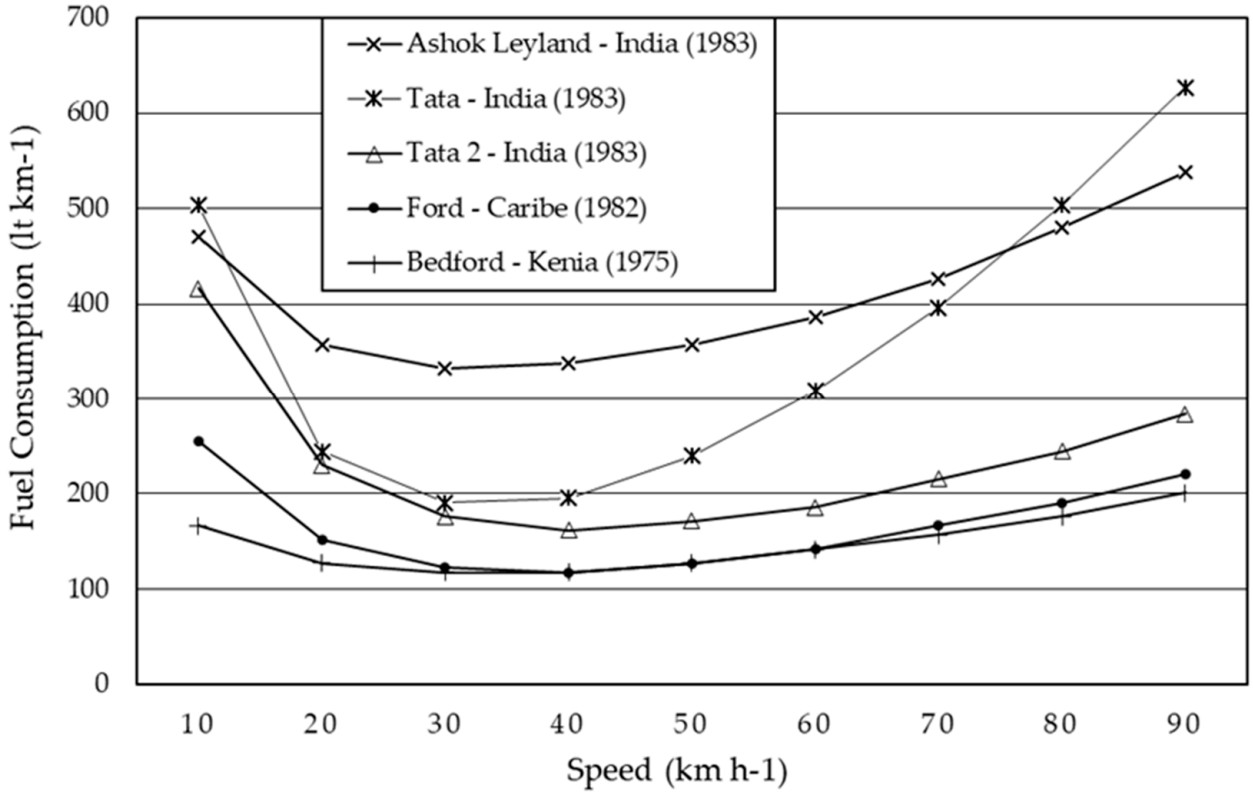

**Figure 5.** Vehicle speed vs. fuel consumption. Redrawn from Altamira [9].

Several studies underlying the HDM-4 fuel consumption model have been conducted in newer vehicles [21] of the 1980s [9,16]. Using such vehicles can influence fuel consumption until parts break-in and vehicle functioning stabilizes. The further lack of inclusion of current, technologically advanced vehicles [9] has raised concern regarding the accuracy of the HDM-4 model. This implies a need for model calibration and adaptation to the site of application [37].

The HDM-4 does not consider articulated trucks with six axles, which comprise an important part of the current vehicle fleet for freight transportation [8]. Therefore, this paper aims to close this gap regarding the knowledge on this topic. The HDM-4 model establishes that fuel consumption is proportional to the engine's total power requirement, which considers three components [18]:

1.  Traction power, which is required to counteract forces opposing the movement.
2.  Engine drag, which is required to counteract the internal engine drag (or friction).
3.  Accessory power, which is required to move vehicle accessories such as fans, power steering, air conditioning, alternator, and so on.

The way that this proportionality is mathematically expressed, in its simple form, is presented in Equation (3) [37]:

$$\text{IFC} = \text{MAX}\,(\alpha,\, \varepsilon \text{Ptot}) = \text{MAX}\,(\alpha,\, \varepsilon \text{Pengaccs} + \varepsilon \text{Ptr}), \tag{3}$$

where

IFC = instantaneous fuel consumption (mL s$^{-1}$);
$\alpha$ = idle fuel consumption (mL s$^{-1}$);
$\varepsilon$ = efficiency factor fuel power (mL kW$^{-1}$ s$^{-1}$);
Ptot = total power requirements (kW);
Pengaccs = total engine and accessories power (kW);
Ptr = total traction power requirements (kW).

Fuel consumption measures or associated models (e.g., those developed with multiple linear regression techniques) are necessary to calibrate the model [37]. As observed, fuel consumption is influenced by diverse variables that jointly, simultaneously, and permanently change values in vehicle operation. A fuel consumption study is usually performed independently for each variable, with the others controlled as defined values, which are then combined; for example, for a single vehicle, speed, freight amount, and so on [8]. Details on fuel consumption models are available in [3,8,9,11,21,36].

There is no validation evidenced for fuel consumption of the proposed models, using the real amount of freight transported by the truck [21,39,40]. This has been simplified, in some studies, by visually identifying empty, half-loaded, or fully loaded vehicle [21,22,39,40]; full load or empty conditions [9,39]; or only using fully loaded trucks [9]. The load amount or vehicle weight variable was analyzed in the present research, which can be considered a significant contribution to knowledge in this field. Furthermore, we considered a truck for which there is no evidence of research having been conducted: an articulated truck with six axles, classified as C3S3 in Colombia, with a total maximum weight up to 52 t [8].

The variables considered in this study were vehicle weight (main element, for the reasons stated above), road slope (no studies considering slope over 6% were found), and vehicle speed (common in all other known studies and models).

## 3. Research Methodology

In this section, we describe the methodology used to assess fuel consumption in trucks. The general methodology of this study consisted of a literature review, followed by planning and conducting field tests to determine the fuel consumption of trucks, according to the load and other defined variables. Then, the information was reviewed, refined, and processed in order to propose models for the estimation of fuel consumption, as well

as making comparisons with the results obtained by the HDM-4 model. This general methodology is shown in a flowchart in Figure 6.

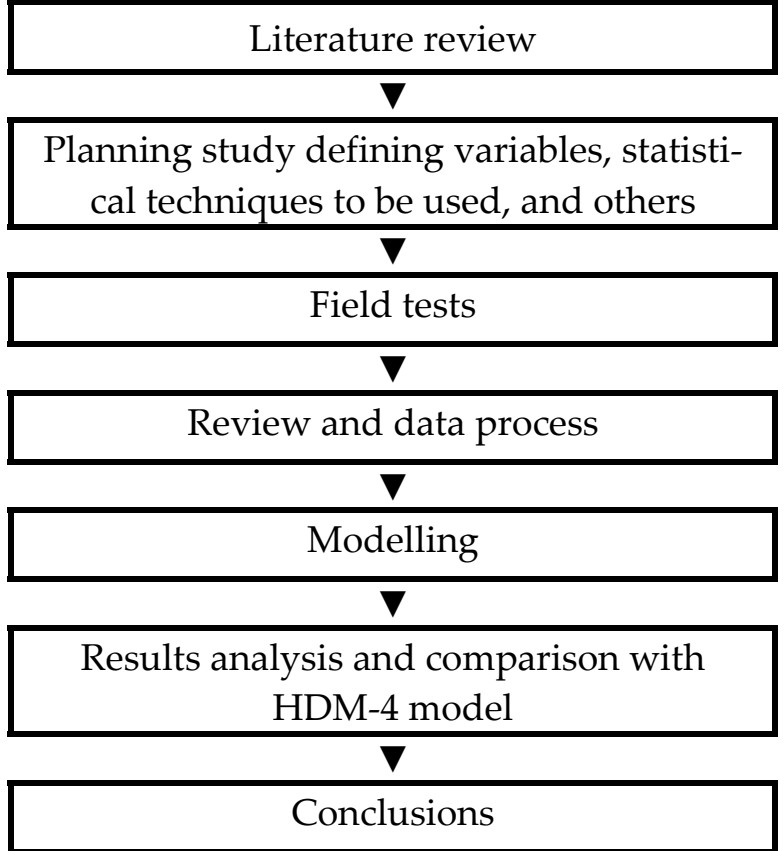

**Figure 6.** General methodology flowchart.

In the experimental part of the research, it was first necessary to identify the equipment and instrumentation used to obtain the information in the field tasks, the amount of data and the manner through which to obtain them, the vehicles to be used, and the roads travelled and days and hours to do them, among other factors, in order to ensure good conditions for the research. The resources used in this investigation were as follows:

- Equipment for measuring fuel consumption in vehicles in motion: Electronic equipment such as the onboard computer and information capture module (datalink), coupled to vehicles with an electronic engine (usually for model or year post-2007). In detail, data about fuel consumption was captured using an On-Board Computer (OBC) Mix Telematics—FM3306 (brand—reference); this OBC has a global positioning system (GPS) incorporated; in addition, it is used a datalink with the J1939 protocol, which can be used in electronic engines as the trucks used have in order to transfer and storage data about fuel consumption and positioning. An image of this OBC and Datalink is shown in Figure 7. The OBC is connected by technical authorized people to the electronic module of the vehicle, and it is stored behind the vehicle console of instruments, as shown in Figure 8.
- Vehicle location and speed identification: Additional to the on-board computer was a GPS Garmin—GPSmap® 60CSx (brand—model), which has good accuracy; it was connected to an external antenna to improve signal reception. In Figure 7, we can see an image of the GPS and external antenna used.

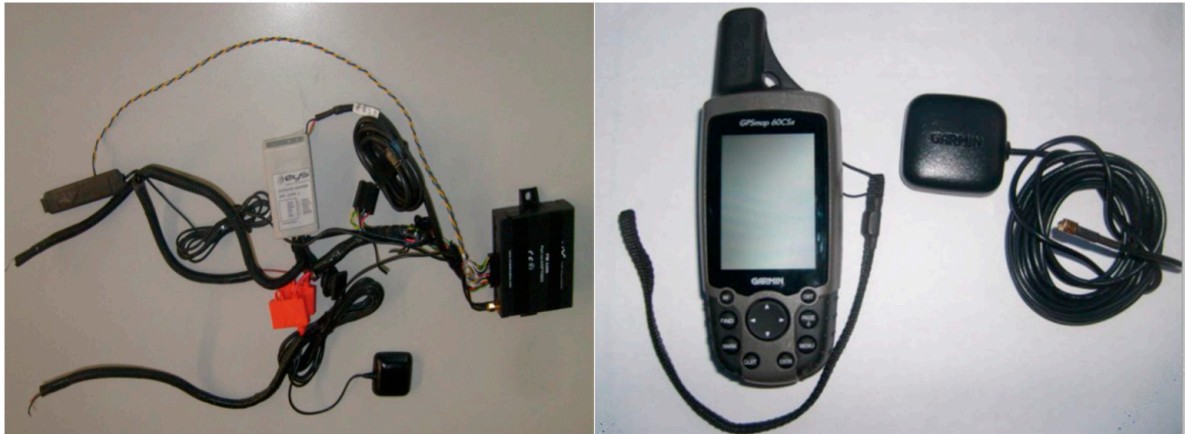

On-board computer and datalink GPS

**Figure 7.** On-board computer and GPS used in the research.

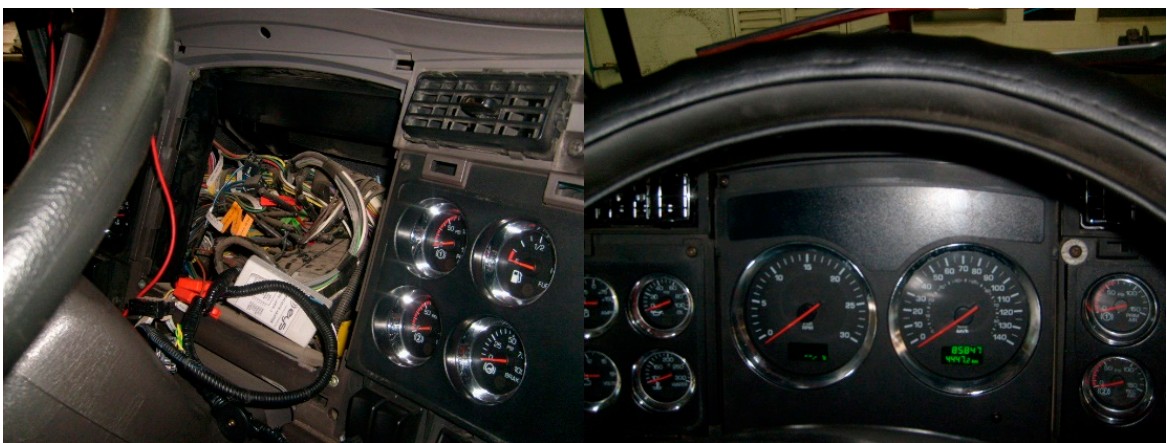

During connection After connection

**Figure 8.** During and after connection of the OBC.

The OBC and GPS were used making their clocks synchronized to obtain data and their processing.

- Trucks used: Rigid-type trucks with three axles, and articulated trucks with six axles (C3 and C3S3 as the official classification in Colombia, respectively), with engine powers of 350 HP (10,800 cc) and 400 HP (15,000 cc), respectively (see Figure 9). These types of trucks were considered essential to the research, considering their market share, future trends in vehicle fleets, recent model (year), technology, and good operating condition and maintenance. The trucks used were checked and, if necessary, technically tuned up before the field test in order to ensure their adequate functioning. All of these characteristics served to ensure the validity of the research for several years.
- Road sections for field testing: A double-lane road and a section with high longitudinal slope (up to 8%) were considered, which allowed for control of vehicle speed and to have several slopes for analysis. The altitude of the road was 1600 m above sea level. Altitude was not considered in this research as an analysis variable, but it can be considered in other studies, as altitude is one of the factors that affects air density and, therefore, the mixture of fuel with air for combustion [17], thus being possibly important in fuel consumption. The road sections were identified before the field test by visual inspection, where their characteristics were those that the research needed; the next step for each one was to make a technical measurement of the length and

slope using topographic methods and surveying instruments; at the end, each road section selected to conduct field tests were those where each one had a constant slope (no variation into each section). Thus, each road section had a unique slope, a different slope from the others. Those road sections selected had slopes up to 7% and lengths ranged from 200 to 400 m. The maximum slope in the research was 7% due to higher slopes having shorter lengths.

- Vehicle speed control: During the experiment, the trucks arrived at each section at the speed needed, controlled using cruise control function of the truck when the truck driver was driving in the segment. Cruise control is a system that can be switched on to select a speed at which the vehicle will continue and to maintain it without the use of the accelerator pedal. The driver has to drive to obtain the desired speed according to the experiment design, then switch on the system and not use the accelerator pedal again until the speed has to be changed for any reason; for this, the driver has to be experienced.

- Trucks drivers: Appropriate drivers were selected, and only one driver was used for each truck for the entire field test. The concept of appropriate drivers was used to describe drivers who drive a specific truck type in the right way, due to their familiarity with the truck type.

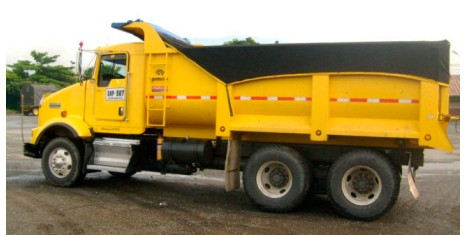
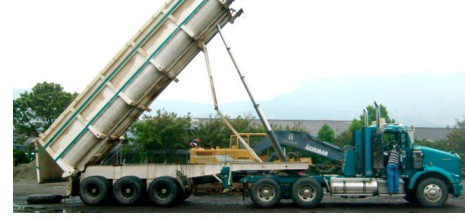

3-axles: C3-type truck               6-axles: C3S3-type truck

**Figure 9.** Trucks used in the research.

To conduct the analysis, we used the statistical methods of experimental design and factorial design to obtain the required amount of data. These techniques optimize testing [41] and save costs without degrading research quality [41,42] and the statistical significance to obtain models. Two types of tests were conducted, according to the results expected. The fuel consumption model in Figures 3 and 5 show curved behavior for speeds up to 45 km h$^{-1}$, while for higher speeds, the behavior tends to become linear. Therefore, for high slopes and lower speed, a greater number of levels was considered for the parameters, in order to identify the curvature effect [41,42], using a $3^K$ factorial design; for low slopes that enable higher speeds, the number of levels for the parameters was lower, as the curvature effect was not considered to be important, and $2^K$ factorial was used. The variation factors and levels used in the experiment are shown in Table 1.

**Table 1.** Factors and levels selected for experimental design.

| Factor | Value | Levels | |
| --- | --- | --- | --- |
| | | $3^K$ **Design** | $2^K$ **Design** |
| Fuel | Diesel (ACPM) | 1 | 1 |
| Type of pavement | Flexible with asphalt | 1 | 1 |
| Pavement condition | Dry and low IRI (<4) | 1 | 1 |
| Traffic flow type | Free flow | 1 | 1 |
| Truck | Rigid and articulated | 2 | 2 |
| Longitudinal slope (%) | Positive, 7% maximum | 3 | 2 |
| Load weight (t) | Up to legal capacity | 3 | 2 |
| Speed (km h$^{-1}$) | Maximum posted | 3 | 2 |

A free flow pattern was selected for two reasons: This is the expected and normal flow pattern in inter-city roads, and this pattern allows the vehicle to be driven at the desired speed, which was one of the factors to be analyzed.

Factorial is the number of combinations of factors (or parameters) and their variation levels; it increases with amount of them, especially with levels. To design the factorial, it is necessary to know factors and their variation levels; factors are the variables of the experiment and levels are values to be considered for each factor in the experiment.

In a mathematical way, the factorial is defined as $\prod \text{Level}^{\text{Factors}}$ ($\prod$ is the symbol for product operator).

There were four factors with one variation level (invariable operating conditions during testing: fuel type, type of pavement, pavement condition, and traffic flow type); one factor (truck) with two levels; and three factors with three or two levels of variation (longitudinal slope of road, weight, and speed of trucks). Then, the factorial design for each case was as follows:

Factorial for $3^K = \prod \text{Level}^{\text{Factors}} = 1^4 \times 2^1 \times 3^3 = 54$;

Factorial for $2^K = \prod \text{Level}^{\text{Factors}} = 1^4 \times 2^4 = 16$.

For each truck, there were 27 data points in the $3^K$ factorial design, and 8 data points in the $2^K$ factorial design. Each datum corresponds to a consumption value, according to the experiment conditions. Three replicas were defined for each test, which was considered sufficient to control means and errors. Thus, 81 data points were required for each truck: 27 defined for each factorial design $3^K$; while the $2^K$ design required 24 data points (8 for each replica).

The level values for the defined factors are presented in Table 2. The road slopes were defined according to the geometric alignment of the road using the ascending way—these slopes were 0.6%, 2.0%, 5.2%, and 7.0% (2.0% were tested for both $2^K$ and $3^K$ factorial designs), and the vehicle weights considered full capacity, medium, or empty. To obtain the maximum operating speed, the trucks were tested at their maximum capacity weight on the higher-sloped road; moreover, low speeds used in tests were selected, considering that we must have at least two more values for them, according to the methodology used in the research, other speeds of trucks observed on the road, and the capability to maintain them at a controlled level.

**Table 2.** Factors and level values in the experimental design.

| Truck Type | Factor | Design $3^K$ | | Design $2^K$ | |
|---|---|---|---|---|---|
| | | Levels | Values | Levels | Values |
| 3 and 6 axles (C3 and C3S3) | Road slope (%) | 3 | 7.0–5.2–2.0 | 2 | 2.0–0.6 |
| 3 axles (C3) | Total vehicle weight (t) | 3 | 27.99–17.18–11.72 | 2 | 27.99–11.72 |
| | Vehicle speed (km h$^{-1}$) | 3 | 45–35–25 | 2 | 70–50 |
| 6 axles (C3S3) | Total vehicle weight (t) | 3 | 51.95–26.28–19.09 | 2 | 51.95–19.09 |
| | Vehicle speed (km h$^{-1}$) | 3 | 35–30–25 | 2 | 60–40 |

Having identified the road stretches, vehicles, and instrumentation, the tests were carried out following the procedure shown in Figure 10 for each truck. In the process shown in Figure 10, it should be understood that truck weight was measured using a calibrated weighing machine before the start of any tour; at the beginning of and during each tour, the free flow traffic condition was checked through communication by cell phone with people on the road who reported on traffic conditions. Future studies may apply another methodology, such as the use of an artificial neural network using historical data to predict traffic flow [43]. During each tour, the fuel consumption was captured by the on-board computer on all road sections and longitudinal slopes considered (0.6%, 2.0%, 5.2%, and 7.0%), and the speed data were captured by both the on-board computer and GPS and stored in those devices. After each tour, the speed data were checked to see if

they were equal in the on-board computer and GPS, and if they were expected according to experiment design; if not, the tour had to be conducted again.

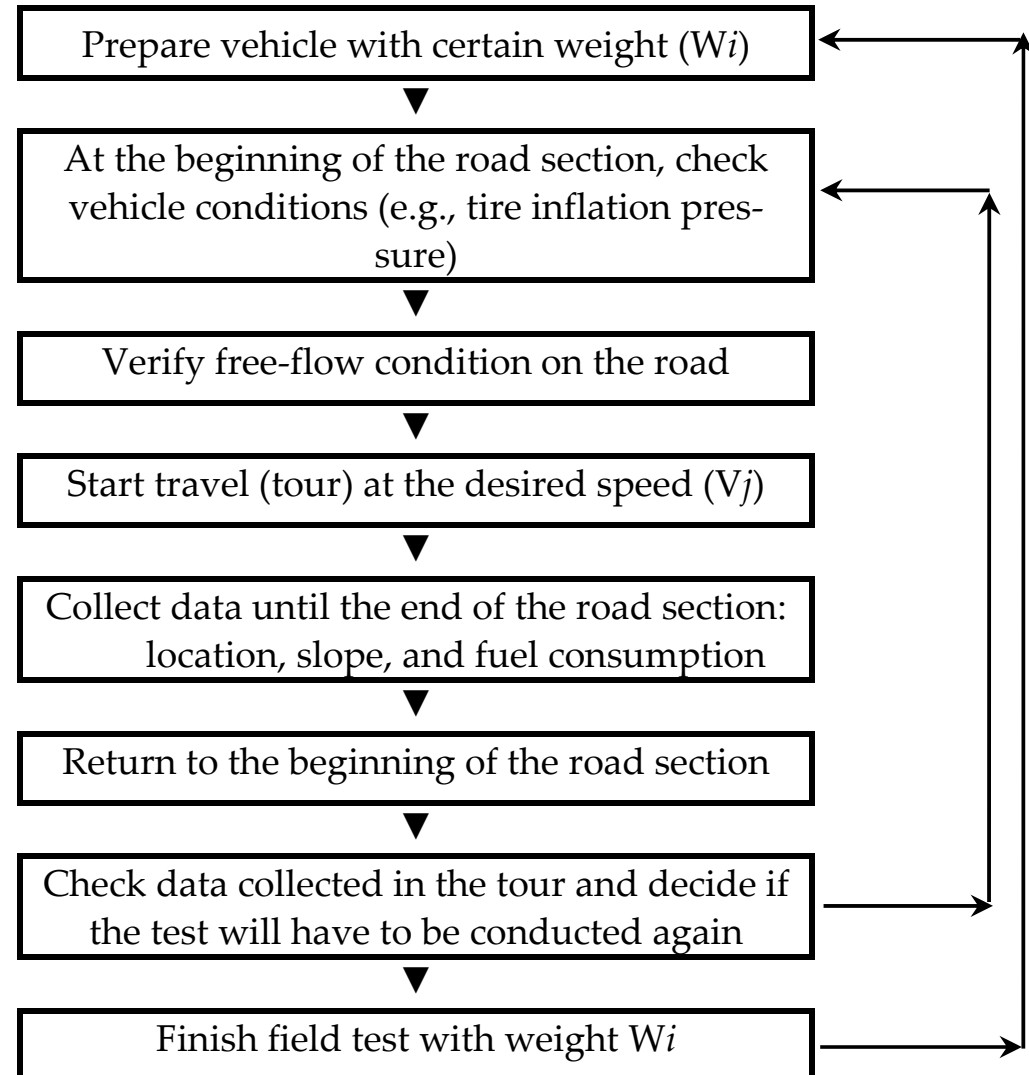

**Figure 10.** Process for conducting field tests.

At the end of each tour, and having right data, the truck had to change its speed to start a new tour, according to the experimental design for the research. When all speeds had been tested for any weight, the amount of freight had to be modified, in accordance with the experimental design; after which, we restarted the field test at the new weight.

At the end of the field work, the data needed to be organized and processed for data analysis regarding fuel consumption, which was the object of this research.

The OBC and GPS were used with their clocks synchronized to obtain data and their processing, and thus when the vehicle was in one of the road sections selected according to GPS data, the fuel consumption data were obtained from the registered OBC data at the same time (clock). Figure 11 shows screenshots from GPS and OBC processing data in one of the road sections selected.

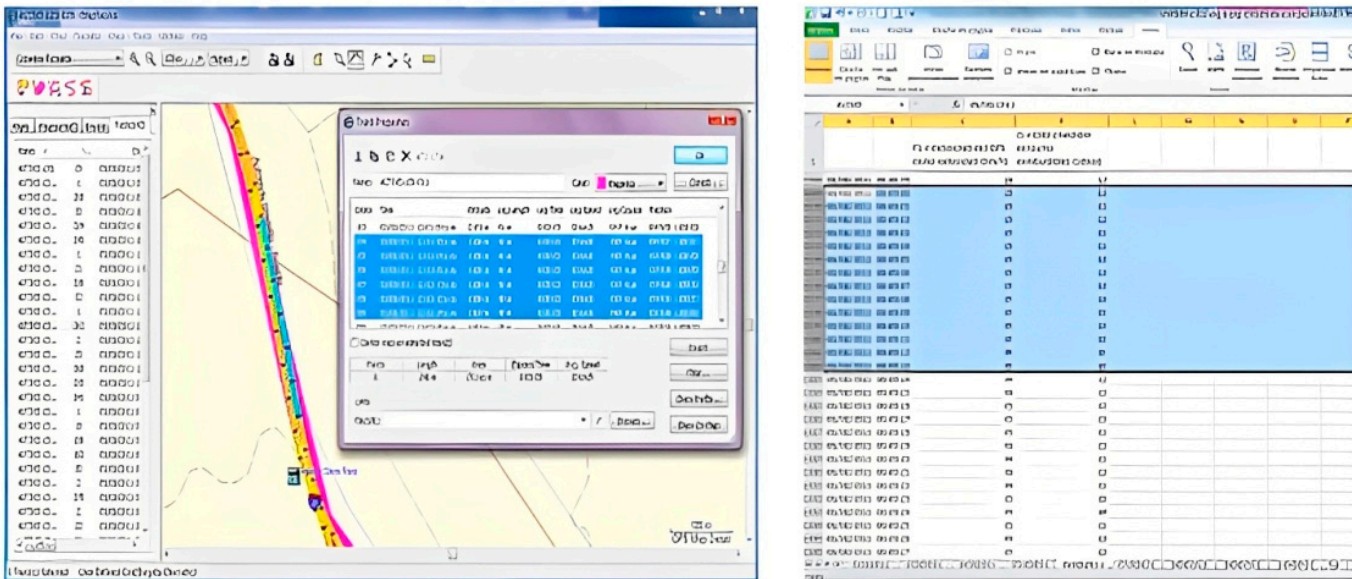

**Figure 11.** Screenshots from GPS and OBC in processing data.

## 4. Results and Analysis

The *Minitab*® software is considered suitable for statistical analysis, such as the factorial experiment design type [41]. Thus, this program was used for the statistical analysis of results, in order to validate the data, identify relationships between the factors studied, and obtain mathematical models that correlates them, as well as determine the validity of the models. Linear and non-linear regressions were carried out for data analysis and to obtain the models considering factor interactions.

Methodological analysis of factorial designs was conducted through analysis of variance (ANOVA) in order to identify the factors and significant interactions. All variables (factors) and their interactions were considered at the beginning of the statistical analysis, where the continuous ANOVA analysis indicated which of them were significantly important to be included in the final models. Table 3 shows the fuel consumption models determined (Equations (4)–(7)), their application range, and their validating statistics with a confidence level of 95% for variables and their interactions. We considered *p*-values less than 0.05 to ensure model validity (the *p*-values obtained included 0.04, 0.009, 0.008, 0.001, and less).

The fuel consumption behavior obtained from the determined models was according other studies conducted on the topic of fuel consumption that show a "U" shape [16,21], suggesting that there is a speed in which the fuel consumption is lower; thus, for high slopes, the speed is 35 km h$^{-1}$ for a 6-axle truck and 45 km h$^{-1}$ for a 3-axle truck.

According to the models and their analysis, we reached the following general observations.

The results indicated that fuel consumption rose when road slope increased, as expected, but this increasing was not proportional. This effect was larger under high-vehicle-weight conditions. Fuel consumption was similar at high slopes (>5%) for low-weight vehicles, as obtained using all models proposed in this paper. Increased fuel consumption due to overall vehicle weight was demonstrated, as accentuated under high longitudinal slope conditions. For example, for a six-axle truck on a 7% longitudinal slope, the ratio of the fuel consumption for a fully loaded (52 t) compared to an empty vehicle (19 t) was 2.92 (consumption at 52 t/consumption at 19 t), while the weight ratio was 2.74 (52 t/19 t). For a 5% slope, the fuel consumption ratio was 2.73, and, for a 2% slope, it was 2.29. Therefore, the consumption ratio was higher than the weight ratio. Results like these can be seen in Table 4, which provides the relative comparison between weight and consumption for different slopes of the road, according to the models obtained for the articulated truck.

**Table 3.** Obtained fuel consumption models.

| Vehicle Type | Mathematical Model | |
| --- | --- | --- |
| | **Fuel Consumption** | |
| C3 | $C = -8.00992W + 105.635S - 3.64516V + 15.2035W \times S - 20.5096S^2 - 0.0270028W^2 \times S^2$<br>Range factors: W: 11.7–28 t; S: 2–7%; V: 25–45 km h$^{-1}$<br>Regression coefficients: $R^2 = 0.9987$; $R^2$adj = 0.9986; $R^2$pro = 0.9985 | (4) |
| | $C = 14.595W + 38.8019S + 2.47673V - 0.207869W \times V - 1.24498S \times V + 0.135391W \times S \times V$<br>Range factors: W: 11.7–28 t; S: 0.6–2%; V: 50–70 km h$^{-1}$<br>Regression coefficients: $R^2 = 0.9998$; $R^2$adj = 0.9997; $R^2$pro = 0.9997 | (5) |
| C3S3 | $C = -49.3166W + 30.2423V + 20.6906W \times S + 0.355453W^2 - 24.9639S^2 - 0.514948V^2 - 0.021823W^2 \times S^2$<br>Range factors: W: 19–52 t; S: 2–7%; V: 25–35 km h$^{-1}$<br>Regression coefficients: $R^2 = 0.9988$; $R^2$adj = 0.9987; $R^2$pro = 0.9986 | (6) |
| | $C = 11.6719W - 1.79316V + 2.9518\,W \times S + 0.0931592W \times V - 0.136448W^2$<br>Range factors: W: 19–52 t; S: 0.6–2%; V: 40–60 km h$^{-1}$<br>Regression coefficients: $R^2 = 0.9994$; $R^2$adj = 0.9993; $R^2$pro = 0.9991 | (7) |

The "*p*" values are zero for all variables in all equations, except in Equation (7), where "*p*" for V is 0.008. C = consumption (mL km$^{-1}$), W = weight (t), S = slope (%), and V = speed (km h$^{-1}$). $R^2$adj = $R^2$ adjusted, $R^2$pro = $R^2$ prognosticated.

**Table 4.** Relative comparison of consumption by weight—truck C3S3—$3^K$ factorial.

| Weight (t) | Weight Ratio | Consumption Ratio According to Road Slope (S) | | |
| --- | --- | --- | --- | --- |
| | | **S = 7%** | **S = 5%** | **S = 2%** |
| 19 to 26 | 1.37 (26 t/19 t) | 1.57 | 1.41 | 1.10 |
| 26 to 52 | 2.00 (52 t/26 t) | 1.85 | 1.93 | 2.08 |
| 19 to 52 | 2.74 (52 t/19 t) | 2.92 | 2.73 | 2.29 |

The results presented in Table 4 suggest that, at a slope of 5%, the fuel consumption ratio closely matched the vehicle weight ratio. At higher slopes, fuel consumption was proportionally higher, generating inefficiencies in transportation from the perspective of fuel consumption. Results such as this reveal that longitudinal slopes higher than 5% are not appropriate in terms of designing a highway if it will be used by high number of trucks (freight corridors).

On the other hand, it was highlighted that the joint effect of weight and slope had a greater effect for high slopes compared to low ones, and that speed affected larger trucks to a greater extent, revealing a non-linear effect. Furthermore, interactions between the variable (factors) vehicle weight, speed, and road slope studied in this research had a significant influence on fuel consumption. The results satisfy the expectations of the research, and some of them demonstrate that the vehicle weight and longitudinal highway slope were significantly more important than speed alone, which was typically considered the main variable in other studies.

Appling analysis using the statistic named Eta-squared as a measure of effect size, it is possible identify that truck weight and road slope significantly impacted fuel consumption, more than vehicle speed. Table 5 shows the Eta-squared for each truck and experiment conducted.

**Table 5.** Eta-squared.

| Variable | Truck C3S3 | | Truck C3 | |
| --- | --- | --- | --- | --- |
| | **$3^K$** | **$2^K$** | **$3^K$** | **$2^K$** |
| Weight (W) | 0.4448 | 0.8061 | 0.4284 | 0.5150 |
| Slope (S) | 0.4631 | 0.1552 | 0.5121 | 0.3726 |
| Speed (V) | 0.0002 | 0.0063 | 0.0063 | 0.0004 |

The results of Table 5 show that weight explained more than 42.8% (54.9% as mean) and slope more than 15.5% (37.6% as mean) of the variation among the fuel consumption, while speed only explained 0.3% as mean. On the other hand, weight and slope had large effects (Eta-squared more than 14%) and speed had a small effect (Eta-squared less than 1%) in terms of the variation on fuel consumption. This result is according to previous ones in which the weight and slope were more important than speed to explain and estimate fuel consumption.

Further analysis, including rigid trucks, enabled us to conclude that using trucks at their maximum load capacity is convenient, even for high slopes, as consumption per unit weight of load transported is lower when compared to other load levels.

Another analysis showed that for a plain highway (longitudinal slopes up 3%) for a cargo amount up to 16 t, which can be transported by 3-axle or 6-axle trucks, it is recommended that 6-axle trucks be used due to their fuel consumption being more efficient (lower values of mL km$^{-1}$ t$^{-1}$)—this is due to better relationships between vehicle weight and machine power (weight/power) for 6-axle as opposed to 3-axle trucks.

Comparing the results of fuel consumption using the HDM-4 model, we observed that this model over-estimated fuel consumption, but not in a constant manner, instead depending on weight, with great variability according to road slope. No factor uniquely correlating them was found; thus, various correction factors to the values found in the HDM-4 were defined for a better representation of the actual conditions studied; for example, for the articulated truck, this factor ranged from 0.83 to 0.37, while, for the rigid truck, it was between 0.88 and 0.50. A study in the United Kingdom showed that the HDM-4 model predicts fuel consumption with errors up 211% for heavy trucks; this value is equivalent to a factor of 0.47 [14], and in a general way it is up to 200% [10].

## 5. Conclusions

In this research, we presented models that enable the prediction of fuel consumption for certain road and vehicle operation conditions. Their results are reflected in different mathematical models that consider the defined variables (truck weight, road slope, and vehicle speed) and were characterized as being highly reliable, due to their confidence level of 95%, as well as their excellent adjustment to real data, as validated statistically. The behavior of these models was according to previous studies, having a "U" shape, meaning that there is a speed to minimize the fuel consumption.

Furthermore, this research included one truck type that has not been previously researched in terms of fuel consumption, that is, an articulated truck with six axles. This is highlighted as a key contribution to the literature, due to the growing use of this type of truck for freight transportation.

The models obtained enabled identification of the use to be given to trucks, or for the selection of a specific truck type, according to the load amount to be transported and the slope of the road. They permit better control of truck use, economic relationships between stakeholders in the freight transportation field, and evaluating road projects with more certainty regarding this cost component.

Vehicle weight, as a key variable affecting fuel consumption, has not been studied extensively or rigorously; therefore, this research closes a certain gap regarding this topic, as trucks do not always transport the same load. Additionally, the effect of high-longitudinal road slope values has not been previously studied, being another aspect considered in this research. We found that the effects of vehicle weight and longitudinal slopes affected fuel consumption more than vehicle speed on its own.

The results demonstrated that increasing vehicle weight affects in particular the fuel consumption for slopes exceeding 5%. Therefore, it is not recommended that freight transportation corridors include longitudinal slopes in excess of 5%, due to the disproportionately high fuel consumption. In the other hand, for a cargo amount up to 16 t, it is recommended that a 6-axle truck instead of a 3-axle truck be used if the cargo will be trans-

ported on a plain highway (longitudinal slope under 3%) due the normal weight/power for these trucks.

The fuel consumption model developed in this paper enables a better evaluation of infrastructure such as tunnels and viaducts (bridges) on freight corridors. It is convenient to use trucks at their maximum freight capacity (in weight), according to the legal limits established regarding the maximum weight allowed for each vehicle.

The HDM-4 model does not represent fuel consumption adequately for the current vehicle fleet operating on roads. Therefore, fuel consumption models must be updated, for instance, every 10 years, such that they can adapt to vehicle changes due to technological advances and to the energetic improvement of fuels, including the increasing proportion of biofuels and gas.

Finally, we can state with conformity that the results obtained through modelling cannot be taken as valid until they are verified under real conditions. Therefore, adaptation, calibration, and model adjustment based on the prevalent study conditions in a determined place or time is necessary.

It is recommended that knowledge be expanded within the topic of this research by conducting other studies considering various aspects such as other vehicle types, the effect of variables considered constant in this research (e.g., altitude of road section), speed changes or "noise acceleration", road curvature, driving style, and fuel type.

**Author Contributions:** The individual contributions by author are as follows: conceptualization, J.J.P.-H. and I.S.-O.; methodology, J.J.P.-H. and A.A.C.-E.; validation, I.S.-O. and A.A.C.-E.; formal analysis, J.J.P.-H., I.S.-O. and A.A.C.-E.; investigation, J.J.P.-H.; resources, J.J.P.-H.; data curation, J.J.P.-H.; writing—original draft preparation, J.J.P.-H.; writing—review and editing, J.J.P.-H., I.S.-O. and A.A.C.-E.; visualization, J.J.P.-H., I.S.-O. and A.A.C.-E.; supervision, J.J.P.-H., I.S.-O. and A.A.C.-E.; project administration, J.J.P.-H. All authors have read and agreed to the published version of the manuscript.

**Funding:** This research received no external funding.

**Institutional Review Board Statement:** Not applicable.

**Informed Consent Statement:** Not applicable.

**Data Availability Statement:** Not applicable.

**Acknowledgments:** The authors acknowledge the Universidad Nacional de Colombia, at Medellín, for all support given to activities conducted by the authors.

**Conflicts of Interest:** The authors declare no conflict of interest.

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
