# Peer review of "Effects of Road Slope and Vehicle Weight on Truck Fuel Consumption"

_sustainability, doi:10.3390/su15010724_

Round 1

Reviewer 1 Report (Previous Reviewer 3)

I have had the opportunity to review a previous submission of this manuscript. in comparison to the previous one, I can tell that English language has been improved.

However, important flaws are still present in the current text.

1. The authors must define what weight ratio means. For example, the ratio  loaded vehicle weight to unloaded vehicle weight.

2. How drivers ensured that speed remained constant during the experiment has not been explained satisfactorily. The driver's expertise and experience? Cruise control system? Do you have any measure of the accuracy when maintaining the speed?

3. The description of the vertical profile of the sections driven is still meagre. In fact, the new version shows contradictions between the highway profile description (line 302, 8%) and the data (Table 2: 7%, 5.2%, 2% and 0.6%). This information is fundamental to ensure the validity of the experiment.

Author Response

Reviewer 2 Report (New Reviewer)

you must write openly Material and Method

and Result and discussion must be expanded

Author Response

Reviewer 3 Report (New Reviewer)

Reviewer Comments

The authors have written on ‘Effects of road slope and vehicle weight on truck fuel consumptions”. I believe the readers would find the article more interesting if most of the sentences in the introduction were appropriately linked to each other. I suggest the service of a language editor to ensure that the manuscript is free from any form of grammatical error. In addition to previously stated observations, the following suggestions should help improve the quality of the manuscript:

1.     The first two lines of the abstract should be rewritten; the sentences do not connect.

2.     What kind of statistical techniques was used and why?

3.     Can the authors provide a sample of the data used?

4.     Last three lines of the abstract are confusing; the authors should rephrase them.

5.     The first paragraph in the introduction section is stuffed with citations, some of which are unnecessary; the authors should reduce them.

6.     What is the importance of the equation in this research?

7.     The authors should introduce a flowchart at the beginning of their methodology section.

8.     The authors should explain the factorial design at the beginning of page 9

9.     The authors should readjust figure 5

10.  I am not satisfied with the result analysis; some major results are missing. The results and discussion section should go beyond just providing tables and numbers.

11. The authors should update the references, so many old references.

Round 2

Reviewer 1 Report (Previous Reviewer 3)

I still have concerns about the use of the variable grade (slope as designated by the authors). The authors refuse to explain precisely how the slope at each point was obtained, beyond just mentioning "topographic methods". The manuscript must contain at least an example of the vertical profile of the test segments overlapped with consumption values.

This comment, in addtition to those of the other reviewers, must be addressed before the manuscript can be considered suitable for its publication.

Author Response

Reviewer 2 Report (New Reviewer)

not all units in the article have been corrected, even though I mentioned it in the first revision. unity in all units within the article.

Since Figure 1 is a copy and paste, the characters in the picture and the characters of the axis names are different. Figure 2 must be redrawn properly.

Figure 2 is so small and of such poor quality that it cannot be read or understood in any way

Figure 3 is so small and of such poor quality that it cannot be read or understood in any way too.

km h-1

I don't have to correct units one by one in the whole article. You have to scan the whole article and correct the units as I have shown.

i can not see A in equation 3.

what is α   in equation 3?

The following are the corrections I first suggested, but none of them have been made. THERE IS NO ANSWER TO THESE COMMENTS

  a little more specific research results can be included in the summary

·         line 34-37  The sentence is left in the air, it must be supported with figures such as emission and fuel consumption values.

·         Line 47-48… Focused fuel consumption figures should be provided here.

·         Line 48… some factors? what are these factors?

·         Line 51… What are some of these gaps? You have to explain them with references

·         Line 80-85… No real figures on fuel consumption supported by sources

·         Line 95… You said some studies, but you gave only one source. More sources should be added and the results from this source should be displayed here

·         Line 134 … which types of vehicles?

·         Equation 2.. units missing

·          Not lt/1000km………lt 1000km-1

·          Not km/h……………..km h-1

·         Figure 1 is copy-paste from other sorces but quality is too bad for quality journal, so you must draw it again

·         Figure 2 is copy-paste from other sorces but quality is too bad for quality journal, so you must draw it again too

·          Not cent/km…..cent km-1

·         Figure 3 is same too, not quality

·         Where's the "A" in equality 3?

·          Not ml/s..…ml s-1

·         Line 287… All specifications and sensitivity values of the equipment used to measure fuel consumption in vehicles should be given (with brands and models)

·         Line 290.. All features and sensitivity values of the GPS used in vehicles should be given (with brands and models)

·         The pictures of the equipment used in the tooling must be given in the article.

·         how the fuel meter and GPS were connected to the vehicles MUST be given in the article with pictures

·         Result and discussion must be expanded, only 2 sources are used for discussion in this section, this is very insufficient, there is not enough discussion.

Author Response

Reviewer 3 Report (New Reviewer)

Dear Authors,

I am satisfied with the answers to the reviewer's comments.

Round 3

Reviewer 1 Report (Previous Reviewer 3)

The last response to the reviewers confirmed my suspicions about the bias in the prediction of fuel consumption as a function of longitudinal grade. If the test site is a single road section with constant longitudinal grade of 7%, it is not possible to predict fuel consumption for different longitudinal grades other than 7% from the results. Therefore, the study makes no sense anymore.

Author Response

Reviewer 2 Report (New Reviewer)

Thank you for all correction

there is just one point too..not km h-1, it must be km h-1 (example)

check all units in all text and fix them as in the example

In all units, the number should be placed above the letter, not next to it. 

Author Response

This manuscript is a resubmission of an earlier submission. The following is a list of the peer review reports and author responses from that submission.

Round 1

Reviewer 1 Report

1.      I suggest removing “of this study” in line 63. You can write “…next the research methodology…”

2.      Line 57: Change “goverment” to “government”.

3.      Line 157: It is important to write about the meaning of coefficients.

4.      Review the figures descriptions. In figure 1, the authors wrote the name and number. In figure 2 and 3, the authors wrote just the number.

5.      The authors have to put the axel name in the graph in the right position.

6.      Lines 237-238: there is a strong affirmative. Is there some evidence?

7.      Line 246: Is there no study considering high slope values? What is “high slope values”. It is important to exemplify.

8.      Line 255: “Among others”: It is important to cited.

9.      Line 271: “The altitude of the road is 1,600 meters above sea level”: is this variable influence in the study? It is important to clarify.

1.   Line 273: What is “appropriate drivers”? It is important to clarify.

1.   Line 284-285: “…the number of levels for the parameters is lower because the curvature effect is not considered important”. It is important to explain.

1.   Table 1,2,3,4 and Figures 4-5: What is the source?

1.   Table 2: Check the numbers in Total vehicle weight – Last Column: 27.99 - ? and 51.95 - ?

1.   Table 2: The authors have to put the units in the second column.

1.   Lines 414-417: The font size need to be adjusted.

Reviewer 2 Report

This study develops truck fuel use models based on accounting for truck weight, speed, and road slope. The models are calibrated using empirical fuel use rates measured via truck onboard electronics. Overall, the study is interesting since the models have good performance. However, it is not clear how generalizable of these models. If they are vehicle-specific, the contribution of the study is low since other researchers will not be able to use the models directly. Methodologically speaking, the novelty is not clearly stated. For example, it is not clear how others can apply the method in other regions. The authors are encouraged to spend time on positioning the contributions of this work amongst literature. For examples, more discussion is needed. Please also see specific comments below:

Line 32: Please add references.

Line 36: What does it mean to “consume the most fuel”? What is the context here?

Lines 39-41: This is not likely to be true. Can the authors provide more evidence for this claim?

Figure 5: Were the tests carried out in real-world traffic? If so, how was traffic condition accounted for? Vehicle maintenance can affect fuel use. Were the trucks serviced before the tests?

Table 3: Can the authors also quantify Eta2 for each factor? Also, please give sample size. It is important to know which factor explains the most variability in fuel use. The R2 is surprisingly high. Did the authors evaluate potential overfitting? It would be good to also conduct cross-validations (e.g., 5-fold cross validation). How generalizable are these models? Are they vehicle specific? It would be good to comment on whether/how readers can apply these models to other trucks.

In the results section, it is important to show empirical data (i.e., measured truck speed profiles and fuel use rates) before introducing modeling results, so that the readers can verify the validity of empirical data.

Reviewer 3 Report

The manuscript presents research conducted to predict truck fuel consumption as a function of the vehicle weight, speed and highway grade.

After having read the text, I am concerned about a few aspects of experiment and the text drafting.

First, I believe that the manuscript would benefit from English proofread. It would enhance the text readability very much.

The references cited in the text are appropriate.

The methodology and experimental description as well as the results and discussion are meagre, particularly in comparison with introduction and the literature review. The former ones should be expanded according to the following comments.

Little information is provided on how the experiment was conducted.

The authors should provide information about the segments driven during the experiment. Segment lengths, whether grade is constant or variable along the segment. If not constant, How was the grade related to fuel consumption.

Where all possible parameter combinations tested? How many times each?

How many times was the segment driven? Which system was used to maintain constant speed during the trip? If not, how did the authors addressed speed variation in the experiment?

How was the fuel consumption exactly measured? Average on the trip or several instant measurements?

The speed ranges are low. Why were not higher speeds tested?

The authors should provide more details of their hypothesis and process conducting the ANOVA test to determine the variables that significantly contributed to the model.

I consider that the interpretation of the model estimates is not correct because interaction variables are involved. This is an important issue.

The concepts of weight ratio and consumption ratio are not clearly explained in the manuscript.

Reviewer 4 Report

1.     The authors were recommended to revise the abstract section to add more information about the methodology and experimental results for the study.

2.     The last three keywords were recommended to be revised to better demonstrate your study.

3.     The academic contributions were recommended to included in the introduction section.

4.     The sentences from line 85 to 87 were difficult to follow. Please rephrase it. Moreover, please proofread the manuscript to correct out similar problems.

5.     The authors only considered the free flow state for consumption measurement. But, traffic jam state was also recommended to be included in the experiment.

6.     The equations from line 297 to 299 were difficult to understand. Please add more explanations.

7.     The following studies may be properly cited in the study: [1] Optimizing electric bus charging infrastructure considering power matching and seasonality. Transportation Research Part D: Transport and Environment. 100:103057. 2021. [2] Sensing Data Supported Traffic Flow Prediction via Denoising Schemes and ANN: A Comparison, IEEE Sensors Journal, vol. 20, pp. 14317-14328, 2020.

Round 2

Reviewer 2 Report

Thank you for the response. However, the response is not informative. It is not clear what has been changed in the manuscript to address the comments. The validity of the results/models is hard to verify given the comments are not properly addressed.

Reviewer 3 Report

The authors have somewhat addressed most of my comments. However, they did not reflect their rebuttals in the manuscript.

English language has not been sufficiently improved.

Information about driven lengths has not been included in the manuscript.

How drivers ensured that speed remained constant during the experiment has not been included.

A justification for low test speeds is missing.

In order to justify the variables of each model, their p-value should be included as well as the significance level considered. If these results are in accordance with literature, then it should be stated explicitly in the manuscript.

The concept of weight ratio is still missing in the manuscript.

Reviewer 4 Report

my comments have been addressed.

Round 3

Reviewer 2 Report

Thank you for providing the response. However, it is not clear what has been changed to improve the manuscript. It is extremely difficult to search in the track changes to find the answers myself. In the future, it is highly encouraged to provide revisions made in the manuscript in the response so that the reviewer doesn't have to spend time reading the manuscript again. In addition, providing "private messages" to the reviewer is not effective. The reviewer would like to see the comments being addressed directly in the manuscript. 

The reviewer still doesn't see any novelty in this study, given the chances for revision. The major issue is that it is well known weight and slope significantly impact fuel consumption. There is nothing new to design a study for this purpose. The authors claim that "vehicle weight and roadway slopes significantly impact fuel consumption, and more than vehicle speed." However, there is no statistical analysis to support this claim. What the reviewer asked for is to quantify the Eta2 for each of the variables. It is surprising to see that the authors fail to provide such an analysis after two revisions.

Reviewer 3 Report

Once again, I believe the authors have somewhat addressed most of my comments, but not all.

The reader can still find several grammar mistakes in the text, so I seriously doubt that the text had undergone native English proofread.

Information about driven lengths has not been included in the manuscript yet. Second time I notice.

How drivers ensured that speed remained constant during the experiment has not been included yet. Second time I notice.

A clear definition of what weight ratio means has not been included. Second time I notice.

With these flaws the reviewer cannot guarantee that the research has been appropriately carried out, making the manuscript not suitable for publication.